# Different Background: Natural Killer Cell Profiles in Secondary versus Primary Recurrent Pregnancy Loss

**DOI:** 10.3390/jcm10020194

**Published:** 2021-01-07

**Authors:** Laura Strobel, Kilian Vomstein, Christiana Kyvelidou, Susanne Hofer-Tollinger, Katharina Feil, Ruben-Jeremias Kuon, Susanne Ebner, Jakob Troppmair, Bettina Toth

**Affiliations:** 1Department of Gynecological Endocrinology and Reproductive Medicine, Medical University Innsbruck, Anichstrasse 35, 6020 Innsbruck, Austria; laura.strobel@student.i-med.ac.at (L.S.); christiana.kyvelidou@i-med.ac.at (C.K.); susanne.tollinger@tirol-kliniken.at (S.H.-T.); katharina.feil@tirol-kliniken.at (K.F.); bettina.toth@i-med.ac.at (B.T.); 2Department of Gynecological Endocrinology and Fertility Disorders, Ruprecht-Karls University Heidelberg, Im Neuenheimer Feld 440, 69120 Heidelberg, Germany; ruben.kuon@med.uni-heidelberg.de; 3Department of Visceral, Transplant and Thoracic Surgery (VTT), Daniel Swarovski Research Laboratory (DSL), Medical University of Innsbruck (MUI), Innrain 66, 6020 Innsbruck, Austria; susanne.ebner@tirol-kliniken.at (S.E.); jakob.troppmair@i-med.ac.at (J.T.)

**Keywords:** recurrent pregnancy loss, natural killer cells, NKp46, NKG2D, cytotoxicity

## Abstract

(1) Background: Prior studies suggested a significant impact of previous live births on peripheral natural killer cells (pNK) in patients with recurrent pregnancy loss (RPL). Patients with primary RPL (pRPL, no live birth) showed higher numbers of pNK than secondary RPL patients (sRPL, ≥ 1 live birth). (2) Methods: To further determine immunological differences between RPL patients and controls, we analysed pNK subpopulations and activation markers in pRPL (*n* = 47), sRPL (*n* = 24) and controls with previous live birth (sCtrl, *n* = 25) and nullipara (pCtrl, *n* = 60) within a prospective study. Percentages and numbers of CD56^dim^CD16^bright^ cells, subpopulations and activation markers (CD57+, CD62L+, NKG2D+, NKp46+) were measured in non-pregnant RPL patients and *n* = 85 controls (*n* = 60 pCtrl, *n* = 25 sCtrl) in the mid-luteal phase by flow cytometry. (3) Results: Compared to sRPL patients, sCtrls showed higher CD56^+^ and CD56^dim^CD16^bright^ numbers. Further, sRPL patients showed lower numbers of CD56^dim^CD16^bright^NKG2D^+^ and CD56^dim^CD16^bright^NKp46^+^ than sCtrls. (4) Conclusion: We suggest a chronic immune stimulation leading to a lower NK-cell count in sRPL patients with a lower NK cytotoxicity. This underlines the necessity to investigate pNK subpopulations as well as pRPL and sRPL separately to delineate the immune alterations in RPL.

## 1. Introduction

Recurrent pregnancy loss (RPL) is defined as two or more consecutive pregnancy losses from the time of conception until 24 weeks of gestation and affects approximately 1–3% of couples trying to conceive [1,2]. Further, RPL can be differentiated into primary (pRPL) and secondary RPL (sRPL). Women with sRPL have experienced at least one livebirth before the pregnancy losses, while women with pRPL did not. Although there are several established risk factors, including maternal age, endocrine and metabolic disorders, parental chromosomal aberrations, anatomic malformations and antiphospholipid syndrome [3], in more than 50% of women the cause of RPL cannot be identified [4]. Recent diagnostic focuses on immunologic risk factors such as natural killer (NK)-cells in the peripheral blood (pNK-cells) and uterine NK-cells (uNK), regulatory T-cells and dendritic cells [5,6,7,8,9,10,11,12]. NK-cells are characterised by the expression of the surface marker CD56 and can be differentiated into CD56^dim^CD16^bright^ and CD56^bright^CD16^dim^ [13]. CD56^dim^CD16^bright^ with a high expression of CD16 resulting in a high cytotoxic potential are more prevalent in the peripheral blood (pNK) [14]. CD56^bright^CD16^dim^ are the most common cell type in the endometrium during the luteal phase and are, therefore, often referred to as uNK with a low expression of CD16 providing a regulatory activity [15,16]. CD56^dim/bright^CD16^dim/bright^ are a heterogeneous population concerning the expression of cell-surface receptors [17]. Fine-tuned interactions of various activating or inhibitory cell-surface receptors such as NKp46, CD62L, CD57 and NKG2D have been associated with different NK-cell functions [18,19]. 

There is evidence that previous live births impact pNK concentrations reflected by a different immune regulation regarding pRPL and sRPL [20]. We have shown that in pRPL (*n* = 151), significantly higher absolute numbers but not percentages were detected in comparison to sRPL (*n* = 85) [20]. Lately, we confirmed these results in a large, well-defined cohort (pRPL= 393, sRPL = 182), showing significantly higher absolute numbers as well as percentages of pNK in idiopathic pRPL (*n* = 167) compared to idiopathic sRPL (*n* = 81) patients [21]. 

To further elucidate the underlying immunological differences between pRPL and sRPL, we aimed to investigate the composition of pNK subpopulations within a prospective cohort study compared to parous and nulliparous controls.

## 2. Materials and Methods

### 2.1. Study Population

In total, *n* = 71 RPL patients (defined as ≥2 consecutive miscarriages, including *n* = 47 pRPL and *n* = 24 sRPL) and *n* = 85 controls were included in this prospective study between March 2018 and August 2020. The controls consisted of women who already had one or more live births (sCtrl, *n* = 25) and nullipara (pCtrl, *n* = 60). Patients were recruited in our recurrent pregnancy loss unit. Controls were recruited using social media, university-based mailing lists and postings on noticeboards at the university. For the controls, inclusion criteria included: age 18–40 years, without regular medication or hormonal contraception, no former blood transfusion, no allo-sensitization, no autoimmune or haemostatic diseases. Diagnostics were performed in non-pregnant RPL patients and controls. In RPL patients and controls, obstetric and medical histories as well as sociodemographic and lifestyle factors were obtained including age, body-mass-index (BMI), smoking, gravidity, parity, number of miscarriages, number of dilatations and curettages. Non-pregnant RPL patients were routinely screened for (i) anatomical malformations by vaginal ultrasound and/or hysteroscopy; (ii) endocrine dysfunctions (polycystic ovary syndrome according to Rotterdam criteria (2004), hyperprolactinemia, hyperandrogenaemia, corpus luteum insufficiency and thyroidal dysfunctions (hypo-/hyperthyroidism, thyroid autoantibodies); (iii) autoimmune disorders (antinuclear antibodies > 1:160, anticardiolipin antibodies (Immunoglobulin G (IgG) ≥ 10 U/mL, IgM ≥ 5 U/mL), anti-ß2-glycoprotein (IgG ≥ 10 U/mL, IgM ≥ 10 U/mL) or lupus anticoagulant); (iv) deficiencies in coagulation factors (protein C, protein S, factor XII or antithrombin); (v) inherited thrombophilia (mutations in the factor V or prothrombin gene) and (vi) parental chromosomal disorders (numerical aberrations). At least 3 months had to have passed since the last miscarriage before starting diagnostics. Patients with chromosomal abnormalities or autoimmune disorders were not included in this study. Patients or controls with no previous live birth were assigned to the group of pRPL or pCtrl, respectively, whereas patients and controls that had at least one live birth before the miscarriages were assigned to the group of sRPL or sCtrl.

Blood samples of controls were taken in patients and controls in the mid-luteal phase of the menstrual cycle between day 5 and day 8 after the mid-cycle luteinizing hormone (LH) surge. Characteristics of RPL patients are displayed in detail in Table 1.

Signed informed consent was obtained from all participants, allowing analysis of all clinical and laboratory data mentioned in this paper. All procedures performed in studies involving human participants were in accordance with the ethical standards of the institutional and/or national research committee and with the 1964 Helsinki declaration and its later amendments or comparable ethical standards. The study was approved by the Medical University of Innsbruck review board (EK Nr: 1210/2017).

### 2.2. Analysis of Peripheral Lymphocytes and NK-Cell Subsets

5 mL of peripheral blood, anticoagulated with EDTA was collected from healthy controls and RM patients. After blocking with Fc Blocking Reagent (Miltenyi, Bergisch Gladbach, Germany, 130-059-901), peripheral blood cells were incubated with the antibody master mixes containing the following antibodies in various combinations: CD3, CD4, CD14, CD16, CD19, CD45, CD57, CD62L, CD127 (BD); CD56, NKG2D, NKp46; FoxP3; CD25 (refer to Appendix A
Table A1 showing antibodies used, including catalogue numbers). Red blood cells were lysed with the RBC Lysis Buffer (eBioscience, San Diego, CA, United States, 00-4300-54), and samples were analysed after staining with 7AAD (BD, 559925). For the FoxP3 intracellular staining, cells were stained with Fixable Viability Dye (eBioscience, San Diego, CA, United States, 65-0866-14), fixed, permeabilized with the Foxp3/Transcription Factor Staining Buffer Set (eBioscience, San Diego, CA, United States, 00-5523), and incubated with Normal Rat Serum (eBioscience, San Diego, CA, United States, 00-5555-94) prior to the incubation with the FoxP3 antibody. For the calculation of absolute cell numbers, BD Trucount^TM^ Tubes (BD, Becton, Dickinson, Franklin Lakes, NJ, United States, 340334) were used according to the manufacturer’s protocol. Finally, all samples were analysed using a BD LSRFortessa flow cytometer (Becton, Dickinson, Franklin Lakes, NJ, United States). Gating strategy is shown in Appendix A
Figure A1.

### 2.3. Statistics

Statistical analysis was performed using SPSS Version 26 (IBM Corp. Released 2020. IBM SPSS Statistics for Windows, Version 26.0. Armonk, NY, USA: IBM Corp). In case of normally distributed raw data, tested by Shapiro–Wilk normality test, Student’s t-test was used to compare two groups. If variables were not normally distributed, Mann–Whitney-U test was used. In case of homogeneity of variance, tested by Levene test, 1-way analysis of variance (ANOVA) was used for multiple group comparisons even if variables were not normally distributed justified with the central limit theorem. Kruskal–Wallis non-parametric test was applied if homogeneity of variance was missing. If significant results were obtained, post hoc analysis using Gabriel and Dunn–Bonferroni was performed to correct for multiple comparison. For all statistical tests performed, *p* < 0.05 was considered statistically significant.

## 3. Results

### 3.1. Study Population

Number of miscarriages, BMI and luteal phase progesterone levels did not differ between the subgroups of RPL patients. Gravidity (and parity) of patients were significantly higher in sRPL versus pRPL patients. PCtrl were significantly younger than all other groups and showed a lower BMI than pRPL and sRPL. (Table 1)

### 3.2. Peripheral Lymphocyte Subpopulation in Controls

Immune diagnostics in pCtrl and sCtrl are shown in Table 2. PCtrl showed significantly lower absolute numbers, but not percentages of CD56^bright^CD16^dim^ cells (mean ± SD per µL: 15.9 ± 6.3 vs. 21.4 ± 9.6 *p* = 0.003; Table 2). No significant differences between pCtrls and sCtrls were detected in CD56^+^, CD56^dim^CD16^bright^ or CD56^+^CD3^+^ NKT-cells. 

### 3.3. Peripheral Lymphocytes and Subsets in Patients and Controls

Peripheral lymphocytes subpopulations (CD45^+^cells) of RPL patients and controls are shown in Table 2. Compared to sRPL patients, sCtrls showed higher CD56^+^ absolute numbers, but not percentages (mean ± SD per µL: 253.3 ± 161.0 vs. 339.8 ± 147.0 *p* < 0.001; Appendix A
Figure A2a). SCtrls showed significant higher absolute numbers but not percentages of CD56^dim^CD16^bright^ pNK-cells in comparison to sRPL and pRPL patients (mean/µL: 308.8 ± 134.8 vs. 232.2 ± 161.4 (pRPL, *p* = 0.007) vs. 162.6 ± 90.3 (sRPL, *p* < 0.001; Appendix A
Figure A2b). In RPL patients, a trend was noticeable between pRPL and sRPL with lower absolute numbers and percentages in sRPL (CD56^dim^CD16^bright^: mean ± SD per µL: 232.2 ± 161.4 vs. 162.6 ± 90.3; mean ± SD in %: 91.9 ± 4.2 vs. 88.2 ± 6.1). Concerning CD56^dim^CD16^bright^ subsets, sRPL showed lower percentages of CD56^dim^CD16^bright^NKG2D^+^ and CD56^dim^CD16^bright^NKp46^+^ compared to sCtrls (mean ± SD NKG2D^+^ in % 95.3 ± 4.4 vs. 97.2 ± 3.0 *p* = 0.006; mean ± SD NKp46+ in % 56.3 ± 23.4 vs. 81.5 ± 7.5 *p* < 0.001). Similarly, pRPL had a significantly lower percentage of CD56^dim^CD16^bright^NKp46^+^ subsets than pCtrls (mean ± SD in % 63.7 ± 20.0 vs. 77.1 ± 11.7 *p* = 0.001; Appendix A
Figure A3a). 

Concerning CD56^bright^CD16^dim^ cells, no significant differences between groups were detected. However, sCtrls showed higher percentages of CD56^bright^CD16^dim^NKp46^+^ subsets in comparison to pRPL and sRPL patients (mean ± SD in %: 98.2 ± 0.9 vs. 96.3 ± 3.9 (pRPL) *p* = 0.019; 98.2 ± 0.9 vs. 95.6 ± 4.1 (sRPL) *p* = 0.007; Appendix A
Figure A3b).

Further, pRPL patients showed significantly higher percentages of CD4^+^ CD25^+^FoxP3+ cells than pCtrls (mean ± SD in % 2.0 ± 1.0 vs. 1.5 ± 0.7 *p* = 0.035; Table 2). 

## 4. Discussion

So far, diagnostic protocols proposed by current guidelines do not emphasize on a distinction of pRPL and sRPL [2,22,23,24]. However, previous studies of several groups have shown intriguing differences in the composition of clinical and immunological risk factors [20,21,25,26]. Recently we have shown that patients with idiopathic sRPL present higher uNK-cells, but lower pNK-cells compared to women with idiopathic pRPL and suspected a possible abnormal recruitment of NK-cells from peripheral blood to the endometrium [21]. However, due to a lack of a control group in this study, it remained unclear whether the changes in the RPL groups were caused by the physiological antigenic challenge during and after birth in the sRPL patients or representing an immune disorder in these patients. In reproductive immunology, the relatively simple classification of only separating CD56^bright^CD16^dim^ NK-cells with cytokine production (uNK) and CD56^dim^CD16^bright^ NK-cells with cytotoxic activity (peripheral blood) was predominant for a long time [27]. However, more recent studies suggested a more diverse classifications of NK-cells [27,28]. Therefore, we aimed to further differentiate the NK subpopulations and compare these with healthy controls with and without live births. 

In this study, we could show that obviously, giving birth leads to alterations in NK-cell populations. However, in our study cohort, an inverse relation of NK-cell numbers and live birth was present. Compared to sCtrl, sRPL patients showed lower numbers of CD56^dim^CD16^bright^, CD56^dim^CD16^bright^NKG2D^+^ and CD56^dim^CD16^bright^NKp46^+^. This finding is in line with a study by Shakar et al., which demonstrated a higher NK activity in pRPL compared to sRPL patients [26]. The NKG2D receptor and natural cytotoxicity receptors including (NKp46, NKp30 and NKp44) are considered the major activation receptors involved in NK cytotoxicity [29,30]. Patients with spontaneous pregnancy loss displayed higher proportions of NKp44 and NKp46 on CD56^dim^CD16^bright^ decidual NK-cells [31]. In 2006, a study including *n* = 24 RPL patients and *n* = 13 healthy controls showed a lower expression of NKp46 in RPL patients suggesting a dysregulation of NK cytotoxicity [32]. However, this study did not differentiate pRPL and sRPL patients. It is important to note that in our study, lower numbers of NKp46^+^ and NKG2D^+^ cells were not only present in RPL patients, but sCTRL showed significantly more of these cells than all other groups. These findings condense into the assumption of a different immune profile and aetiology in pRPL and sRPL, possibly due to foetal microchimeric cells transferred during pregnancy and most importantly during delivery [33]. Foetal progenitor cells have been shown to persist for up to 27 years [34]. A chronic immune stimulation could, therefore, lead to a lower NK-cell count in sRPL patients with a lower NK cytotoxicity. In a previous study of our group on T-cell subsets and cytokine assays in RPL patients, we suggested an immunological disorder in RPL patients similar to T-cell exhaustion in HIV and cancer, with increased HLA-DR but decreased CD25 expression on CD3+ T-cells, which were less responsive to mitogens [35]. We suspect a similar immune exhaustion in the NK subset, which is possibly more pronounced in sRPL patients due to a higher prevalence of obstetrical complications during the previous deliveries, such as placental abruptions or preeclampsia [36,37,38]. One might speculate that this NK-cell exhaustion could lead to a higher susceptibility for diseases such as cancer or common infections such as herpesvirus in sRPL patients. However, a large cohort study of over 28,000 patients with cancer and 283,294 matched controls showed no association between RPL and cancer. The subgroup of pRPL patients had a borderline significant association to cancer (OR 1.27 (1.04–1.56)), whereas sRPL was not associated with a higher risk for cancer [39]. Other studies that questioned an association of RPL and cancer did not differentiate between pRPL and sRPL patients [40]. To our knowledge, there are no further data on herpesvirus or other common infections in being more prevalent in pRPL or sRPL, as there are in NK-deficient patients [41]. 

A systematic review and meta-analysis suggested that women with sRPL were more likely to obtain a potential beneficial effect from using intravenous immunoglobulin (IVIG) [42]. In contrast, if immunized using allogenic leukocyte immunization, pRPL patients had a higher live birth rate than sRPL patients [43]. 

As a limitation of our study, it must be noted that the pRPL and sRPL as well as the pCTRL and sCTRL groups showed significant differences in age and BMI, which could possibly confound our findings. However, previously, we could not show a significant influence of clinical parameters such as BMI, age, time of last miscarriage or progesterone levels on pNK and uNK-cell numbers in RPL patients [20]. The influence of BMI and age on lymphocyte count is discussed controversially and studies did not compare small differences in weight and BMI in RPL patients versus controls [44,45,46]. Further, the sample size of our study is too small to extrapolate the results of our study to a broader population. Future studies will have to focus on NK-cell subpopulations in larger cohorts, possibly in a multi-centre setup.

In conclusion, our study underlines the necessity to investigate NK-cell subpopulations to further delineate the immune alterations in RPL patients. A distinction of pRPL and sRPL as well as pCtrl and sCtrl could reduce the heterogeneity of research on RPL in general [47]. Future research could focus on humoral immunity in these patients as well, to further decipher the underlying immune mechanisms of the cellular alterations shown in this study.

## Figures and Tables

**Table 1 jcm-10-00194-t001:** Characteristics of patients with pRPL, sRPL and controls.

Characteristics	pRPL (*n* = 46)	sRPL (*n* = 24)	pCtrl (*n* = 60)	sCtrl (*n* = 25)	*p*
Age ^a^	35.5 ± 5.4	35.2 ± 4.4	24.8 ± 3.1	33.4 ± 6.4	<0.001
BMI ^a^	25.7 ± 4.9	25.3 ± 5.1	22.0 ± 3.5	23.0 ± 3.7	<0.001
Gravidity ^b^	3(0/8)	4(3/8)	0	1(1/3)	<0.001
Parity ^b^	0(0/1)	1(1/3)	0	1(1/3)	<0.001
No. of miscarriages ^b^	3(2/8)	3(2/6)	0	0	<0.001

Data presented as ^a^ mean ± SD, ^b^ median (min/max). Statistical analysis by ANOVA or Kruskal–Wallis test whenever applicable. Significant *p*-values (*p* < 0.05) are marked in bold. BMI = body mass index; No. of miscarriages = number of miscarriages; pCtrls = controls without pregnancy; pRPL = primary recurrent pregnancy loss; sCtrls = controls with previous pregnancy; sRPL = secondary recurrent pregnancy loss.

**Table 2 jcm-10-00194-t002:** Immune diagnostics in patients with pRPL, sRPL and controls without (pCtrl) and with (sCtrl) previous pregnancy.

Peripheral Lymphocytes Subpopulations		Unit	pRPL (*n* = 46)	sRPL (*n* = 25)	pCtrl (*n* = 60)	sCtrl (*n* = 25)	*p*
	CD45^+^	/µL	2734 ± 721	2358 ± 699	3067 ± 738	2897 ± 751	0.002 ^3^
*CD45*	CD56^+^	/µL %	253.3 ± 161.0 9.5 ± 4.7	184.7 ± 88.8 7.9 ± 2.9	282.5 ± 165.8 9.4 ± 5.0	339.8 ± 147.0 10.4 ± 3.8	0.003 ^4,5^ 0.327
	CD56^bright^CD16^dim^	/µL %	16.7 ± 6.9 7.8 ± 4.7	15.8 ± 6.5 9.2 ± 4.5	15.9 ± 6.3 6.4 ± 3.3	21.4 ± 9.6 6.2 ± 2.4	0.114 0.063
	CD56^dim^CD16^bright^	/µL %	232.2 ± 161.4 91.9 ± 4.2	162.6 ± 90.3 88.2 ± 6.1	254.6 ± 152.6 91.9 ± 4.2	308.8 ± 134.8 90.7 ± 4.6	0.003 ^3,4,5^ 0.122
	CD56^+^CD3^+^NKT	/µL %	61.9 ± 48.3 2.3 ± 1.8	65.2 ± 55.4 2.1 ± 1.4	73.0 ± 51.6 2.6 ± 2.0	84.6 ± 67.4 2.9 ± 2.2	0.496 0.772
	CD4^+^CD25^+^FoxP3^+^	%	2.0 ± 1.0	2.0 ± 1.0	1.5 ± 0.7	1.7 ± 0.5	0.035 ^2^
*CD56^bright^CD16^dim^*	CD57^+^	%	2.5 ± 4.5	1.6 ± 2.1	2.3 ± 5.2	1.3 ± 3.2	0.434
	CD62L^+^	%	91.9 ± 7.8	94.2 ± 5.1	94.8 ± 5.4	94.4 ± 5.4	0.111
	NKG2D^+^	%	92.3 ± 7.6	92.5 ± 6.9	91.2 ± 8.7	96.0 ± 4.8	0.234
	NKp46^+^	%	96.3 ± 3.9	95.6 ± 4.1	97.2 ± 1.6	98.2 ± 0.9	0.030 ^1,4,5^
*CD56^dim^CD16b^right^*	CD57^+^	%	34.3 ± 16.3	41.5 ± 15.5	32 ± 14.9	32.3 ± 15.2	0.077
	CD62L^+^	%	27.3 ± 11.2	28.8 ± 12.1	29.6 ± 11.3	31.8 ± 12.3	0.483
	NKG2D^+^	%	95.6 ± 3.7	95.3 ± 4.4	96.9 ± 2.2	97.2 ± 3.0	0.029 ^4,5^
	NKp46^+^	%	63.7 ± 20.0	56.3 ± 23.4	77.1 ± 11.7	81.5 ± 7.5	<0.001 ^2,3,4,5^

Data presented as mean ± SD. Statistical analysis by ANOVA or Kruskal–Wallis test whenever applicable. Significant *p*-values (*p* < 0.05) are marked in bold. ^1^ marks a sig. difference between pCtrl and sCtrl. ^2^ marks a sig. difference between pCtrl and pRPL. ^3^ marks a difference between pCtrl and sRPL. ^4^ marks a sig. difference between sCtrl and pRPL. ^5^ marks a sig. difference between sCtrl and sRPL. NKT = natural killer T-cells; pCtrls = controls without pregnancy; pRPL = primary recurrent pregnancy loss; sCtrls = controls with previous pregnancy; sRPL = secondary recurrent pregnancy loss.

## Data Availability

The data presented in this study are available on request from the corresponding author. The data are not publicly available due to ethical reasons.

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
