# Peer review of "Different Background: Natural Killer Cell Profiles in Secondary versus Primary Recurrent Pregnancy Loss"

_jcm, 2021, doi:10.3390/jcm10020194_

Round 1

Reviewer 1 Report

This paper set out to elucidate the underlying immunological differences between primary and secondary recurrent pregnancy loss by investigating the composition of peripheral natural killer cell popluations in women suffering from recurrent pregnancy loss. The manuscript is very well written and the objective is clear. The “n” is a little small, especially compared to the previously published cohorts described by the authors in the Introduction. However, the addition of analyzing NK cell subpopulations was a novel touch that greatly improved the story. A few comments:

Line 40: “focusses” should be focuses

Lines 95-96: Could the authors provide catalog information for the antibodies used? This could perhaps be a supplemental table, but would be useful information nonetheless.

Table 1: It is less of a problem that the pCtrl group has a significantly younger mean age and lower BMI. However, the difference in mean BMIs between the sCtrl and the pRPL and sRPL is problematic. There are several papers describing how obesity can dysregulate NK cell function. At the very least, this point should be addressed in the Discussion.

Table 2&3: It’s fine that the data is presented in tables but the density plots for the flow cytometry from which the data was derived should at least be included in the supplementary.

Reviewer 2 Report

The purpose of this manuscript is to “further elucidate the underlying immunological differences between pRPL and sRPL, we aimed to investigate the composition of pNK-subpopulations within a prospective cohort study compared to parous and nulliparous controls.”  This was a prospective, cohort study. 

  1. How did the authors arrive at their ‘n’?  Did they perform an ‘a priori’ sample size determination or some other method of sample size determination?
  2. How did the authors recruit subjects and controls for this study? Were these consecutive patients with RPL and controls that they saw in their clinic?  Some other method of recruitment?
  3. Could the authors list inclusion and exclusion criteria? They note a number of screening tests that they performed in subjects with RPL.  Did all subjects with RPL included in this trial have all of the tests for RPL?  Were the subjects in this trial those with unexplained RPL?  Or did they also include subjects with chromosomal anomalies, uterine malformations, endocrine anomalies, etc? 
  4. Could they list inclusion and exclusion criteria for the controls?
  5. In the analysis of peripheral lymphocytes and NK cell subsets, how many replicates were run of each sample?
  6. The authors note in the discussion that “higher prevalence of obstetrical complications.” Which obstetrical complications?

Reviewer 3 Report

  The paper authored by Strobel et al attempted to demonstrate that compared to sRPL patients, sCtrls exhibited higher CD56+ and CD56dimCD16bright numbers. In addition, sRPL patients showed lower numbers of CD56dimCD16brightNKG2D+ and CD56dimCD16brightNKp46+ than sCtrls.   Although the topic is of interest there are several deficiencies. Many problems exist in the manuscript. For example NKG2D is written incorrectly. The manuscript has to be edited entirely.   Some specific comments are below     - The manuscript is full of repetitions.   

-Table 1 which is characteristics of patients can be moved to M&M or can be presented as subject information in a separate section. 

-Please specify the surface expressions of NKT cells in Table II.

-pCtrl and sCTrl results of NK and NKT subsets in Table III are similar to Table II. To avoid repetition, please delete table 2, or select to present the results in one table. Fig. 1 and Fig. 2 are also presenting the results shown in table III. It is just shown in a different way, therefore, it can easily be moved to the supplementary file. 

-Please be consistent with labeling and/or description,  table III specified CD45+ cells which was not clarified in table II although the same numbers are presented for each immune subset. 

Reviewer 4 Report

Abstract and discussion: The Authors argue that the NK cell phenotype suggests lower cytotoxicity in blood NK cells of sRPL patients. This is reasonable but Authors should test this in straightforward functional assays measuring cytotoxicity in NK cell subsets (the CD107a assay and intracellular IFN-g would be appropriate). 

Abstract and through the paper: It is NKG2D, not NKGD2

Table III: The label 'Peripheral lymphocytes..' is covered up

Table III. It makes little sense that absolute numbers of CD56+ and CD56dim are increased in cCtrl but % are not. Is it the case that CD45+ total numbers are also increased in sCtrl? The Authors should probably add CD45+ numbers for all groups. 

Speculation on immune exhaustion or chronic activation: this is a reasonable speculation, however the Authors should also discuss the possibility that this might be expected to be associated to increased susceptibility to herpesvirus infections or maybe even malignancy in a way at least close to what is experienced by NK-deficient patients? 

Round 2

Reviewer 2 Report

The authors have responded to queries.  Thank you for submitting this interesting manuscript/revision.